# Influences of Multicenter Bonding and Interstitial Elements on Twinned γ-TiAl Crystal

**DOI:** 10.3390/ma13092016

**Published:** 2020-04-25

**Authors:** Zehang Fu, Jinkai Wang, Hao Wang, Xiaogang Lu, Yanlin He, Ying Chen

**Affiliations:** 1State Key Laboratory of Advanced Special Steels, Shanghai Key Laboratory of Advanced Ferrometallurgy, School of Materials Science and Engineering, Shanghai University, 99 Shangda Road, BaoShan District, Shanghai 200444, China; fuzehang@shu.edu.cn (Z.F.); ceasea@i.shu.edu.cn (J.W.); xglu@shu.edu.cn (X.L.); ylhe@staff.shu.edu.cn (Y.H.); 2Materials Genome Institute, Shanghai University, 99 Shangda Road, BaoShan District, Shanghai 200444, China; 3Department of Nanomechanics, School of Engineering, Tohoku University, 6-6-01 Aramakiaoba, Aoba-ku Sendai 980-8579, Japan; ying@rift.mech.tohoku.ac.jp

**Keywords:** γ-TiAl, twin boundary, first-principles, local energy, three-center bond

## Abstract

The bonding properties of the twin boundary in polysynthetic twinned γ-TiAl crystal and the effect of interstitial alloy elements on it are investigated by first principles. Among the three different kinds of interface relationships in the γ/γ interface, the proportion of true twin boundaries is the highest because it has the lowest interfacial energy, the reason for which is discussed by local energy and three-center bond. The presence of the interstitial atoms C, N, H, and O induces the competition for domination between their affinity to host atoms and three-center bonds, which eventually influences the values of unstable stacking fault energy (USFE) and intrinsic stacking fault energy (ISFE). The relative importance of different bonding with different alloy elements is clarified based on the analysis of local energy combined with Electron Localization Function (ELF) and Quantum Theory of Atoms in Molecules (QTAIM) schemes.

## 1. Introduction

TiAl alloys have the advantages of low density, good high-temperature strength, excellent oxidation resistance, and excellent creep resistance, showing promising potential in aerospace materials. However, the defects of TiAl alloys, such as difficult hot working and poor plasticity at room temperature, still limit its practical application in engineering [1,2].

The existence of Ti d-Al p-oriented covalent bonds in TiAl leads to brittleness of the material [3]. By adding an appropriate amount of transition elements such as V, Mn, Nb, and W to replace Al atoms in TiAl, the interaction between Ti and transition alloy elements can be enhanced and the ductility can be improved [3]. By adding interstitial solid solution elements B, C, and N, the directionality of the nearest Ti–Ti bond can be enhanced, while the interstitial elements H and O can weaken the directionality of the Ti–Ti bond [4]. Therefore, B, C, and N can improve the ductility of TiAl, while H and O cannot [4]. By adding C, the thickness of the γ and α_2_ lamellar structure can be refined at the same time and the yield strength and creep resistance of the material can be improved [5].

However, the improvement of room temperature ductility of TiAl alloys is relatively limited by alloying. Therefore, a lot of research has been done on its microstructure [6]. The interface in the microstructure has a very important influence on the plasticity of materials. The plastic deformation ability of materials can be adjusted by changing the interface spacing and the number of interfaces in the structure and the interface spacing [7,8]. Compared with large-angle grain boundaries, coherent twin boundaries (TB) have attracted much attention due to their better strengthening and toughening effects [9,10]. The existence of TB can interact with dislocations, hinder the movement of dislocations, and play a similar strengthening role to traditional grain boundaries. In addition, nano-twin can absorb a large number of dislocations near TB, thereby improving the work hardening ability and plasticity of the material [9,11].

Chen [12] manufactured a Ti-45Al-8Nb alloy with a well-aligned 0°-oriented polysynthetic twinned (PST) single crystal. Its yield strength reached 930–1035 MPa at room temperature and the elongation reached 6.3%–7.6%. This is due to the existence of the twinning process at the intermediate stage that the ductility of the Ti-45Al-8Nb alloy is improved. In addition, research by Lu et al. [9,10] also showed that the presence of nano-twin in Cu can increase strength without reducing ductility. In TiAl alloys with a single γ phase, the main deformation mechanism is nanotwinning [13]. When the temperature is higher than 800 °C, dislocations will pass through TB in the single γ phase, which will cause TB to migrate and improve ductility [13]. The ductility of a single γ phase is mainly due to the nano-twin throughout the γ grains.

The γ/γ interface in the PST crystal has a very important influence on the mechanical properties of the lamellar structure [14]. On the one hand, the existence of the γ/γ interface results in a thinner γ slat, which results in better mechanical properties of the structure; however, the γ/γ interface will become a source of cracks [15,16]. According to relevant experiments, it has been proved that the thickness of the lamellar structure will affect the ductility, creep resistance, and fracture toughness of the material [17]. Increasing the number of interfaces can to a certain extent improve ductility and creep resistance [18]. However, the variation of bonding properties on twin γ-TiAl interfaces in presence of alloy elements has not yet clarified.

In this paper, the variation of bonding properties on the γ-TiAl twin boundary is analyzed by different methods in the presence of interstitial elements. The article is divided into four parts. The introduction is in the first section, followed by presentation of the theoretical methods and models in Section 2. Section 3 presents the results and discussion. A summary is provided in Section 4.

## 2. Theoretical Method and Model

### 2.1. Model and Calculation Parameters

The first principles calculation is performed using the VASP software package developed by the University of Vienna using the plane wave pseudopotential method based on density functional theory (DFT) [19]. The projector augmented wave (PAW) method is used with the Perdew–Burke–Ernzerhof (PBE) generalized gradient approximation (GGA) functional in order to obtain the energy and stress densities. The energy convergence criterion of the electronic self-consistency is chosen as 10^−4^ eV for all the calculations. A convergence tests guided the choice of 500 eV for the cutoff energy of all the calculations. A 15 × 17 × 1 Monkhorst–Pack grid of k-points is adopted for the twin model in Figure 1.

γ-TiAl has an L1_0_ crystal structure with c/a ≈ 1.02. According to the related research [20], γ-TiAl is often used as the face-centered-cubic with c/a = 1, which makes it easier to describe the orientation relationship between the crystals and the effect on the stacking fault energy can be almost ignored. Two unit cells are spliced through the (111) Z plane to obtain the true twin model of γ-TiAl with the directions of the coordinate axes X = $\left[1\bar{1}0\right]$ and Y = 1/2$\left[11\bar{2}\right]$. The twin model constructed in this calculation contains 48 atoms, whose size is enough to accommodate interstitial alloy elements by testing different sizes.

The interfacial energy of twin is defined relative to the total energy possessed by the bulk crystal. The energy of the bulk crystal model is calculated by constructing a bulk model with the same size and number of atoms as in the twin model. The interface energy calculation formula is (1)$$\gamma =\frac{E_{tot}-E_{bulk}}{2A}$$
where *E_tot_*, *E_bulk_*, and *A* are the total energy of the twin model, the energy of the bulk crystal, and the interface area, respectively.

There are six symmetrically different octahedral interstitial positions as shown in Figure 1**,** which can be divided into Al-rich octahedral interstitial positions (I1–I3) and Ti-rich octahedral interstitial positions (I4–I6). Totally, 12 different tetrahedral interstitial positions exist in the model. The formation energy formula of interstitial atom is defined as (2)$$\Delta E\left(X\right)=\frac{E_{T+X}-N1*E_{Ti}-N2*E_{Al}-N3*E_{X}}{N1+N2+N3}$$
where *E_T+X_*, *E_Ti_*, *E_Al_*, and *E_X_* are the total energy of the model in presence of interstitial atoms, the energy of a single Ti atom in the bulk α-Ti model, the energy of a single Al atom in the bulk fcc-Al model, and the energy of one interstitial atom in the molecule of simple substance, respectively. *N*1, *N*2, and *N*3 are the numbers of Ti, Al, and interstitial atoms in the model, respectively.

### 2.2. Theoretical Methods

Quantum Material Simulator (QMAS), a code similar to VASP, is utilized for the energy density calculations [21,22], which has been successfully applied to the analysis and calculation of grain boundary (GB) tensile test [23,24]. The energy density $\epsilon \left(\vec{r}\right)$ is defined as (3)$$E_{tot}=\int_{V}^{}\epsilon \left(\vec{r}\right)d\vec{r}$$
where *E_tot_* is the total energy with cell-integrated or cell-averaged quantities conventionally obtained by the plane-wave DFT method [25]. The local energy *E_i_* of the *i*-th local region with a volume *V_i_* in the supercell is given by integration of each density as (4)$$E_{i}=\int_{V_{i}}^{}\epsilon \left(\vec{r}\right)d\vec{r}$$

However, the local energy suffers from the gauge-dependent problem of the kinetic terms, i.e., the non-uniqueness or ambiguity for the selection of the symmetric or asymmetric forms of the kinetic terms. The practical approach to removal of the present gauge dependency is to define the local region *V_i_* where the difference between the symmetric and asymmetric forms is integrated to be zero [22]. An atomic region, *V_i_*, can be attained by Bader partitioning of the valence-charge density distribution into each atomic region, where the gradient component of the charge density normal to the region surface is zero [26], and therefore the difference between the symmetric and asymmetric forms of the kinetic energy density is integrated to be zero [27]. The reliable Yu–Trinkle algorithm is adopted to perform accurate Bader partitioning for the charge density data on the fast Fourier transformation (FFT) mesh points [19,28].

CRITIC2, which is a program for analyzing quantum-mechanical atomic and molecular interactions in periodic solids [29,30], applies localization of critical points (CPs), gradient path tracing, and basin integration to scalar field in Quantum Theory of Atoms in Molecules (QTAIM) analysis [31]. The core of the QTAIM is the electron density ρ(**r**) distribution, the first derivative of which at a point in space is 0 (∇ρ(**r**) = 0) if this point is called CP. Among four stable CPs: nuclear critical points (NCP) means that ρ(**r**) reaches a local maximum and bond critical points (BCP) means that ρ(**r**) reaches a maximum value in two perpendicular directions to and a minimum value in the parallel direction to the bond path connecting a BCP and an NCP. Non-nuclear maxima (NNM) is a CP with maxima in three directions at a point in space instead of NCP [32]. By comparing ∇^2^ρ(**r**), energy density H_t_, electron kinetic energy density G, electron potential energy density V, and |V|/G and H_t_/ρ at CPs, the topological properties of the molecules can be obtained, and the characteristics between the atoms can be analyzed. If ∇^2^ρ(**r**) is negative, the atoms tend to be covalent, and if it is positive, it is closed-shell [31]. |V|/G is greater than 2 for covalent interaction, less than 1 for non-covalent interaction, and between 1 and 2 for partial covalent interaction [33]. When |V|/G is greater than 2, bond degree (BD) is equal to H_t_/ρ, and BD is a negative value, which can indicate the covalent action intensity. The larger the absolute value of BD, the stronger the covalent action.

## 3. Results and Discussion

### 3.1. Structure Analysis of Twin Model

#### 3.1.1. Pure Twin Model

After the optimization of the γ-TiAl unit cell, the lattice constant a = b = c = 4.011 Å, α = β = γ = 90° is obtained. There are 12 layers, labeled from Layer1 to Layer12 in the twin model shown in Figure 1. The calculated twin interfacial energy is 57.31 mJ/m^2^, which is very close to 60 mJ/m^2^ calculated by Fu [17].

Theoretically, an Electron Localization Function (ELF) [34] value of 0.5 indicates that the Pauli repulsion has the same value as that in a uniform electron gas of the same density, whereas an ELF value smaller (or larger) than 0.5 corresponds a local Pauli repulsion larger (or smaller) than that in a uniform electron gas [35]. In other words, ELF = 0.5 represents the electron–gas like pair probability; ELF = 0 and ELF = 1 represent perfect delocalization and localization, respectively [36]. In general, metallic bonding is supposed to be delocalized; covalent bonding is supposed to be localized [36,37]. Figure 2 is the ELF of the Layer7 and Layer10 in the pure twin model. It can be seen that the ELF_max_ = 0.765 in the regions surrounded by Ti13, Al13, and Ti14 (labeled as Al13-T) and by Ti13, Al14, and Ti14 (labeled as Al14-T) on TB is higher than the ELF_max_ = 0.715 in the regions surrounded by Ti19, Al19, and Ti20 and by Ti20, Al20, and Ti19 on Layer10 in bulk region, even higher than ELF_max_ = 0.718 at BCP on the TB located between Al13 and Al14. Therefore, a stronger interaction in the Al13-T and Al14-T regions is formed and labeled as three-center bonds [38].

Table 1 shows the properties of ELF and QTAIM at NNM of the three-center region and BCP on Layer7 at 0 shear displacement under pure and interstitial alloy element environment. NNM generally exists between two or more atoms like in Al13-T(Al14-T) in pure twin model, while BCP locates between two atoms [32]. Considering |V|/G > 2, H < 0 and ∇^2^ρ(**r**) < 0 of BCP, the interactions between the Al13 and Al14 atoms on TB tend to be Shared shell [31]. Moreover, the absolute value 0.523 of BD at NNM of Al13-T is larger than the absolute value 0.323 of BD at BCP; furthermore, ρ(**r**) and |V|/G are larger, but ∇^2^ρ(**r**) and H_t_ at NNM are smaller than the ones at BCP. Therefore the covalent interaction in Al13-T and Al14-T regions with NNM is stronger and thus more stable than the covalent interaction between Al13 and Al14 atoms [31]. The analysis results by the ELF in Figure 2 are in good agreement with the one of atomic interaction by QTAIM in Table 1.

Figure 3a shows the layer energy Δ*E_L_* summed up by two Ti and two Al atomic energy Δ*E_i_* in each layer and the atomic energy of single Ti and Al atom in each layer in a pure twin model compared to their corresponding atomic energy in γ-TiAl. Due to periodicity and symmetry, there are only four different atomic layers in the pure twin model, and only the layer energies of layers from Layer7 to Layer10 are drawn in Figure 3a It can be found from Figure 3a that the layer energy of TB (Layer7) is much lower than the bulk region (Layer10). This is because of the existence of these two three-center bonds in Figure 3a that the atomic energy of Ti atoms and Al atoms on TB is reduced at the same time, which results eventually in the layer energy of TB being lower than that of the bulk region. Therefore, the analysis results of the ELF in Figure 2, atomic interaction of QTAIM in Table 1 and the local energy data in Figure 3a are consistent.

It is known [17] that the formation of a TB, a special kind of coherent boundary, decreases the total interfacial energy because the excess energy for coherent TB is much smaller than that for conventional high-angle GBs. In order to check the generality of low layer energy of TB, some typical fcc metals are investigated, such as Cu and Al, which reveal quite different mechanical behaviors due to different bonding nature of Cu and Al with a fully-occupied *d* band and *sp* valence electrons, respectively. The stacking fault energy (SFE) of Al is much higher than that of Cu, leading to narrow and wide splitting widths for partial dislocations in Al and Cu, respectively, which result in rather different microstructure evolution and mechanical properties for Al and Cu due to different dislocation mobility [39,40]. The SFE for Al higher than that for Cu can be explained by the covalent nature of Al and its remarkable redistribution of *sp* valence electrons for the atomic-structure change, compared to the isotropic metallic bonding in Cu [41]. A 0.205 eV lower layer energy at TB than in the bulk region is found for the pure Al twin structure and 0.008 eV also for pure Cu twin structure. Obviously, the difference between the pure Al and pure Cu twin structure is due to their aforementioned bonding nature; furthermore, the similarity between pure Al and γ-TiAl twin structure is due to Al *sp* valence electrons. Therefore, twin structure associated with three-center bonds does cause the lower layer energy at TB as well as inherent valence electron property in elements.

#### 3.1.2. Twin Model with C, N, H, and O

An important aspect of the alloying behavior of intermetallic compounds is the choice of solid solution sites for alloying elements, which will have a significant impact on the material properties. After layers size test, a 12-layer structure is found sufficient to accommodate interstitial elements, whose energy difference from 18-layer structure is only 0.0003 eV. Our calculation results and related literatures [42,43] show that the impurity atoms C, N, H, and O tend to occupy the octahedral interstitial positions rather than tetrahedral interstitial positions; therefore, the tetrahedral interstitial positions are not considered in this paper. In the twin model, the existence of TB will affect the stability of interstitial atoms in the octahedral interstitial positions. Table 2 shows the formation energy when interstitial atoms C, N, H, and O are added to different octahedral interstitial positions in the twin model.

It is found that TB affects the stability and formation energy of interstitial atoms at different octahedral interstitial positions. Ti-rich octahedral interstitial positions are more suitable for interstitial atoms compared with Al-rich octahedral interstitial positions, especially in the I4 position (its octahedron is composed of Ti13, Ti14, Ti15, Ti16, Al14, and Al15, see Figure 1**.**) with the lowest total energy, which therefore is selected hereafter.

Only NNM in the Al13-T region remains on Layer7, and NNM in the Al14-T region near C or N disappears when interstitial atom C or N is added, while there are still one NNM in both Al13-T and Al14-T regions on the Layer7 when the interstitial atom is H or O. However, all interstitial atoms whose NNM still exist have a higher ρ(**r**) but a lower ∇^2^ρ(**r**) and H_t_ compared to pure twin model. It is shown in Figure 4 that the ELF_max_ value at the NNM in Al13-T region is higher than that of the pure twin, and the one at the Al14-T region is decreased when C or N is added, whereas the ELF_max_ values in the Al13-T and Al14-T regions on the Layer7 are increased when H or O is added. In addition, compared with the NNM of the pure twin, the BD in Table 1 in the presence of interstitial atom is higher. Therefore, the covalent interaction at the NNM in the twin models with interstitial atom is stronger. The addition of interstitial atom can make the related NNM more stable, which is consistent with the influence of ELF_max_ increasing at the three-center bond.

Figure 3b is the layer energy of the twin model after the interstitial atoms C, N, H, and O are added to the I4. The layer energies of the two atomic layers (Layer7 and Layer8) near the interstitial atom are reduced. Therefore, the existence of interstitial atoms has a stabilizing effect on adjacent layers, which also indicates that interstitial atoms are prone to segregate near TB.

In order to separate the chemical effect (CAE) from the size effect (SAE) of interstitial atoms on atomic energy of host atoms during relaxation in the presence of interstitial atoms, the branch contribution of interstitial atom on the atomic energy of host atoms is shown in Figure 3c,d by gray, which is obtained by removing the interstitial atom after relaxation and then calculating the atomic energy without atomic relaxation (only electronic relaxation). Figure 3c is the single atomic energy of the corresponding Ti atom on Layers (The atomic energy of the two Ti atoms on each layer is the same.). It can be seen that the decrease in the layer energy of the Layer7 and Layer8 layers is mainly due to the reduction in the Ti atomic energy, the chemical effect (CAE) to be exact, especially when the interstitial atom is C, N, and O. The SAE influence in presence of H on Ti atomic energy is close to its on Al14 and Al15 atomic energies on the octahedron, which will be discussed later. The little influence of interstitial atom on Ti atomic energy on Layer6 is reasonable due to a far distance between them. Figure 4e,f is the ELF of the plane where interstitial atom, Ti13, Ti14, Ti15, and Ti16 in octahedron in Figure 1 are located. It can be seen in Figure 4e,f that the electrons between interstitial atoms and the first neighbor Ti atoms (Ti13, Ti14, Ti15, and Ti16 in octahedron in Figure 1) are highly localized, indicating a strong interaction, except that the electrons localization between H and these Ti atoms is relatively low. The ELF analysis in Figure 4e,f is consistent with atomic energy change in Figure 3c.

Figure 3d is the atomic energy of each Al atom. When an interstitial atom is added, the atomic energy of the two Al atoms on each layer is different due to its long-ranged *sp* orbitals [27,44]. By comparing the Al atomic energy in each layer in the pure twin, it can be found that when the C is added, the Al13 atomic energy is more greatly reduced than Al14 on TB. N segregation at I4 site causes Al13 atomic energy decreasing and inversely Al14 atomic energy increasing slightly. Larger SAE effect than CAE on atomic energies of Al11 and Al12 on Layer 6 is similar as Ti atomic energy on Layer6. However, interstitial atoms play different roles in the variation of atomic energy on Layer7 and Layer8, within which the interstitial atoms are besieged in the octahedron. As relatively larger atoms than H and O, SAE on Al14 by C (N) presence overwhelms (almost equals to) CAE while CAE of C and N contributes more or less on Al13, indicating their size influence on Al14 is larger than on Al13 due to Al14 close to interstitial atom. C and N influence on Al15 and Al16 are similar as Al13 and Al14.

It can be found by ELF in Figure 4 that when C or N is added, the degree of electrons localization between C (or N) and Al14 is high, indicating a strong interaction. Moreover, the addition of C makes the ELF_max_ in the Al13-T region increase from 0.765 to 0.774, indicating an increase of the degree of electrons localization in the Al13-T region and thus an enhancement of interaction between Al13, Ti13, and Ti14. N segregation to I4 site causes ELF_max_ in the Al14-T region to change from 0.765 to 0.734 while the ELF_max_ in the Al13-T region increases from 0.765 to 0.781, similar to the influence by C addition. Therefore, the addition of both C and N can enhance the three-center bond of the Al13-T region and Al13 as the second nearest neighbor Al atom of C (N), has the weak interaction on C (N), resulting in the decrease of atomic energy of Al13 comparing to pure twin γ-TiAl. Meanwhile, when C is added, the ELF_max_ in the Al14-T region decreases from 0.765 to 0.735 (NNM disappears eventually), indicating a decrease in the degree of electron localization in the Al14-T region, and thus a weakening of the interaction between Al14, Ti13, and Ti14, which would theoretically result in an increase of Al14 atomic energy.

However, the atomic energy of Al14 in Figure 3d is reduced to −0.0923 eV in the presence of C compared with −0.0669 eV in the pure twin model. Apparently, the interaction between Al14 and the interstitial atom makes sense and the discussion on the relative influence of the interaction and the weakening of the original three-center bond on Al14 is desirable. Assuming that the aimed atom is much more affected by its first neighboring atoms than other atoms, the dropping of Al14 atomic energy implies the interaction between C and Al14 overwhelms the weakening of the three-center bond in the change of the atomic energy of Al14. The interaction between C and Al14 is dominated by C size effect by Figure 3d. On the contrary, the addition of N makes the atomic energy of Al14 slightly increase by only 0.0326 eV compared to the pure twin model, indicating the enhancement effect produced by N and Al14 on Al14 atomic energy is not as high as the weakening effect of the three-center bond.

As for Al15 in the octahedron (note Ti13, Ti14, Ti15, Ti16, Al14, and Al15 in Figure 1), the degree of electrons localization between C (or N) and Al15 in Figure 4i–l increase much, indicating a C (N)-Al15 covalent bond generated compared with the pure twin model. Therefore, in the presence of C or N, Al15 atomic energy is lower than that of the pure twin model, but still higher than Al14 due to three-center bond of Al14 atom. The addition of C or N creates an interaction with Al14, which results in a change in the strength of the two three-center bonds on Layer7.

The increasing ELF_max_ at the Al13-T and Al14-T regions under the effects of H and O on are similar, making surrounding Al atomic energy increase. As shown in Figure 3d, CAE of H and O on surrounding Al atoms overwhelm SAE of them except for O effect on Al16. Due to the further distance between Al13, Al16, and H (or O), the amount of increase in atomic energy of Al13 and Al16 is less than that of Al14 and Al15 in Figure 3d. However, from Figure 4c,d and Table 1, an increase of ELF_max_ in Al14-T region indicates the three-center bond is enhanced, which could lower atomic energy of Al14 and Al15. According to Figure 3d and Figure 4k,l, in spite of the attractive (negative) interaction between H (or O) and Ti [45], the interaction between H (or O) and Al14 (Al15) shows repulsive (positive) interaction, indicating that the weak *affinity* of H or O on them is much higher than that of three-center bond, opposite to the strong affinity effect of C or N. Therefore, the variation of the atomic energy of Al atoms, ELF_max_, and the number of NNM are eventually the opposite for C/N and H/O interstitial elements, giving rise to different effects on plasticity.

The poor plasticity of γ-TiAl is related to the strong covalent interaction between Ti and Al atoms [3]. Based on the aforementioned, the analysis of local energy and ELF indicate that the addition of C and N enhances the plasticity; by contrast, H and O weaken the plasticity of twin γ-TiAl. The conclusion seems to be consistent with the result calculated in bulk-TiAl by Dang [4] and experiment by Yamauchi [46]. However, the properties of alloys and compounds are largely influenced by the alloy elements and microstructures. It can be imagined that the three-center bond due to the twin structure and affinity of interstitials to host atoms contribute competitively to the plasticity in twin γ-TiAl.

### 3.2. Shear along $\left[11\bar{2}\right]$

In order to investigate the plasticity in twin γ-TiAl, the top six layers of the twin model are sheared along $\left[11\bar{2}\right]$ direction and the shear plane is between Layer7 and Layer6. During shearing, atoms are only allowed to relax in the *Z*-axis direction perpendicular to TB. Moreover, the shear of the top seven layers and shear plane between Layer7 and Layer8 is also tested and is found to have largely high unstable stacking fault energy (USFE) and intrinsic stacking fault energy (ISFE). Therefore, the shear between Layer6 and Layer7 is discussed hereafter. It is noted that a combining movement by slip and transition of interstitial atom is recommended [47], which is out of range of this work. According to the energy curve in Figure 5, USFE and ISFE values in Table 3, the presence of all interstitial atoms increase USFE and ISFE of the twin models with an ascendant order of C-H-N-O for USFE and H-C-N-O for ISFE, respectively.

Figure 6 is the ELF of Layer6 and Layer7 under different shearing displacement in the pure twin γ-TiAl. It can be seen from Figure 6a under 0.18 shearing displacement that compared to ELF of TB of the initial model in Figure 2a, the degree of electrons localization on Layer7 changed significantly: the two three-center bonds in the Al13-T and Al14-T regions are weakened to the extent that they even disappear. When interstitial atoms C, N, H, and O are added, the situation is similar. Therefore, the increase in the layer energy of the Layer7 is caused partially by the weakening or even breaking of the three-center bonds during the shear test. From Figure 6b,c, when the shearing displacement amount reaches 0.36, two new three-center bonds are formed on the Layer6, becoming a new TB. This is exactly in accordance with our previous judgment that the lower energy of TB than that of the bulk region is due to the existence of the three-center bonds.

Figure 7 shows the layer energy of Layer7 and Layer6 under different shearing displacement. It can be found that as the shear progresses before ISFE, the tendency of the layer energies of Layer6 and Layer7 for different alloy element environments remains in spite of gradually increasing of layer energies, which is consistent with the initial tendency under different alloy element environments at 0 shearing displacement. In our opinion, USFE and ISFE in shear progress are strongly correlative to the initial configuration at 0 shearing displacement, which in this study are three-center bond due to the twin structure and affinity of interstitials to host atoms. It is two enhancing three-center bonds (Al13-T and Al14-T) due to O alloy atom insertion instead of its weak affinity that cause its highest USFE and ISFE. On the contrary, although N addition makes one three-center bond vanish, its affinity to host atoms makes sense nonetheless, which results in its second high USFE and ISFE.

## 4. Conclusions

In this paper, ELF, QTAIM, and the local energy are used to study the γ-TiAl twin model and its shear process in the presence of interstitial atoms. It is concluded that a kind of three-center bond is formed on TB, which contributes to the lower layer energy at the TB layer compared to that at the bulk region and are even more stable than BCP between Al atoms; in the presence of the interstitial atoms C, N, H, and O, the three-center bonds they influence compete for dominating stability at TB with their affinity to host atoms, which consists of a size effect and a chemical effect. Two enhancing three-center bonds by O atom overwhelm its affinity and the affinity of N atom makes more sense conversely. The analysis of configuration of initial structure influenced by interstitial atoms is consist with the variation of ISFE and USFE in presence of them during shear test; the local energy of one atom can be an *accordatura*, based on which the relative importance of different bonding is clear combined with ELF and QTAIM schemes. Furthermore, the stability of one atom is quantitatively measured by local energy, which is more applicable than ELF and QTAIM schemes in presence of alloy atoms and in process of shear test.

## Figures and Tables

**Figure 1 materials-13-02016-f001:**
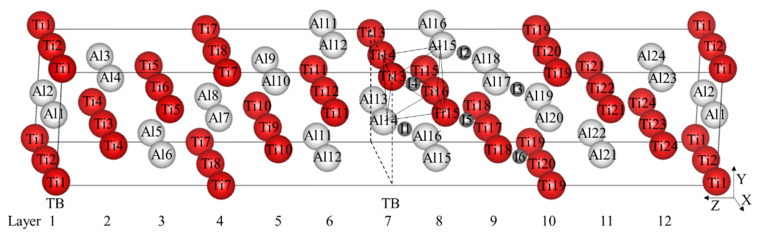
(Color on line) γ-TiAl twin model and 6 octahedral interstitials. The red larger spheres are Ti atoms, the white larger spheres are Al atoms, the gray smaller spheres are octahedral interstitial positions, and the labels in the spheres represent the corresponding atomic numbers or interstitial numbers. I1–I3 represent Al-rich octahedral interstitial positions and I4–I6 represent those that are Ti-rich. The label TB and dotted lines represent twin boundary. Ti13, Ti14, Ti15, Ti16, Al14, and Al15 form an octahedron besieging I4 interstitial site.

**Figure 2 materials-13-02016-f002:**
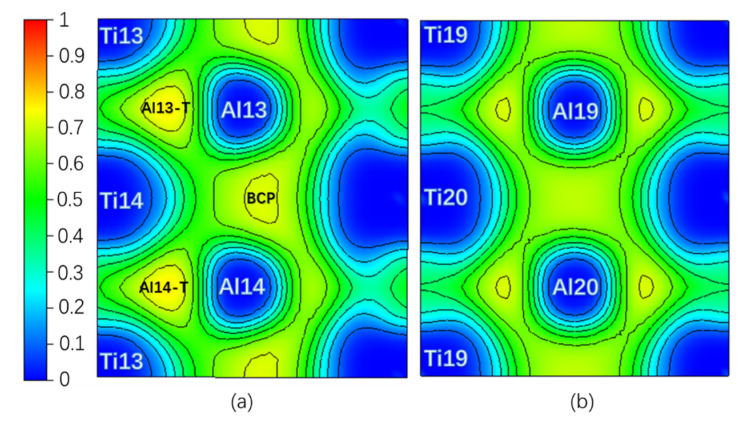
(Color on line.) (**a**) Electron Localization Function (ELF) on Layer7, the maximum value is 0.765; (**b**) ELF on Layer10, the maximum value is 0.715 in pure twin. Al13-T and Al4-T are three-center bonds.

**Figure 3 materials-13-02016-f003:**
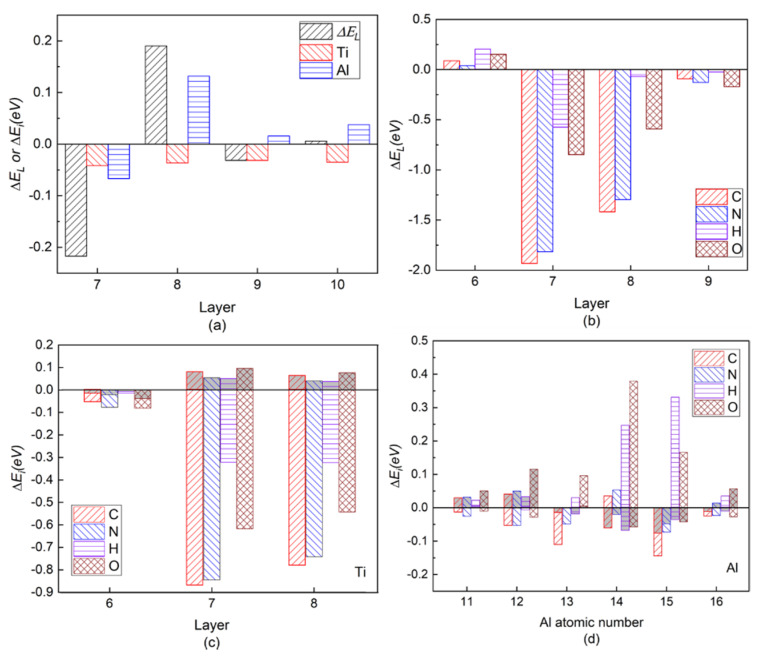
(Color on line.) (**a**) Local energy of pure twin. Symbol “Δ*E_L_*” represents the layer energy (including two Ti and two Al atoms), and symbol “Ti” and symbol “Al” represent the atomic energy of a single Ti atom and a single Al atom in the corresponding atomic layer, respectively. (**b**) Layer energy from Layer6 to Layer9 in the presence of interstitial atom C, N, H, and O in the twin model compared to Ti and Al atomic energies in pure γ-TiAl without twin structure. (**c**) Single Ti atomic energy compared to Ti atomic energy in pure twin model. (**d**) Single Al atomic energy compared to Al atomic energy in pure twin model in layers from Layer6 to Layer8. The gray part in each histogram represents SAE, whereas the residual part represents CAE caused by C/N/H/O insertion.

**Figure 4 materials-13-02016-f004:**
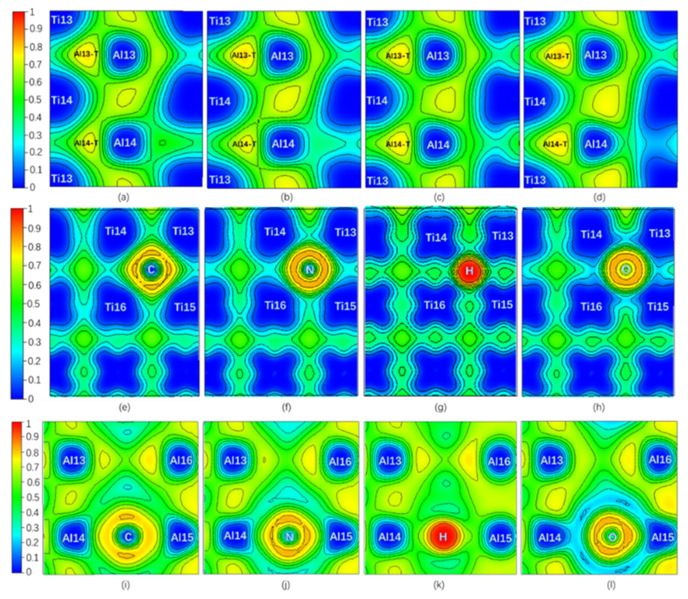
(Color on line.) ELF of twin model in presence of interstitial atom. (**a**) ELF on Layer7 with C. (**b**) ELF on Layer7 with N. (**c**) ELF on Layer7 with H. (**d**) ELF on Layer7 with O at Ti-rich octahedral interstitial positions I4. (**e**) ELF on the plane consisting of Ti13, Ti14, Ti15, Ti16, and C. (**f**) ELF on the plane consisting of Ti13, Ti14, Ti15, Ti16, and N. (**g**) ELF on the plane consisting of Ti13, Ti14, Ti15, Ti16, and H. (**h**) ELF on the plane consisting of Ti13, Ti14, Ti15, Ti16, and O at Ti-rich octahedral interstitial positions I4. (**i**) ELF on the plane consisting of Al13, Al14, and C. (**j**) ELF on the plane consisting of Al13, Al14, and N. (**k**) ELF on the plane consisting of Al13, Al14, and H. (**l**) ELF on the plane consisting of Al13, Al14, and O at Ti-rich octahedral interstitial positions I4.

**Figure 5 materials-13-02016-f005:**
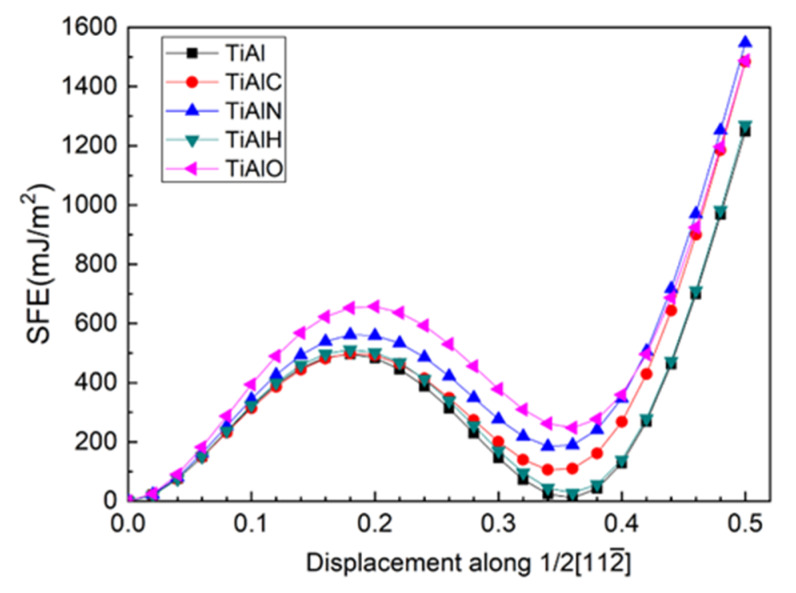
(Color on line.) USFE and ISFE for twin γ-TiAl for slip system (111)/$\left[11\bar{2}\right]$.

**Figure 6 materials-13-02016-f006:**
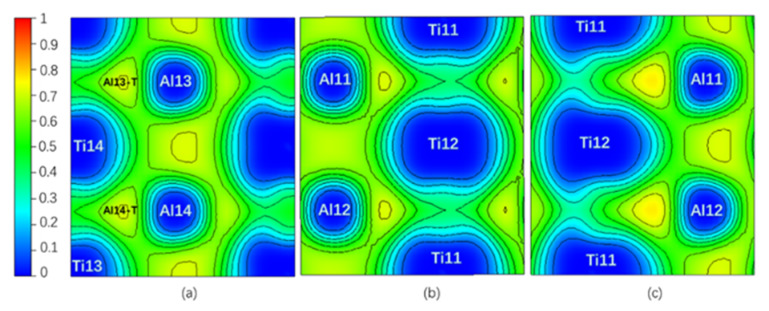
(Color on line.) (**a**) ELF on Layer7 under 0.18 shearing displacement, (**b**) ELF on Layer6 under 0 shearing displacement, and (**c**) ELF on Layer6 under 0.36 shearing displacement in a pure twin γ-TiAl.

**Figure 7 materials-13-02016-f007:**
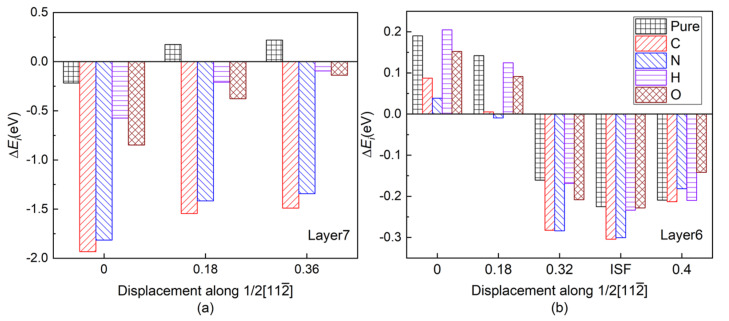
(Color on line.) Layer energy of Layer7 and Layer6 under different shearing displacement (**a**) Layer7 and (**b**) Layer6. The shearing displacement of ISFE is somewhat different for each interstitial atom.

**Table 1 materials-13-02016-t001:** Properties of ELF and Quantum Theory of Atoms in Molecules (QTAIM) at non-nuclear maxima (NNM) of the three-center region and bond critical points (BCP) on Layer7 at 0 shear displacement under pure and interstitial atom environment.

		ELF_max_	ρ(r)(eÅ^−3^)	∇^2^ρ(r)(eÅ^−5^)	G(a.u.)	V(a.u.)	H_t_(a.u.)	|V|/G	H_t_/ρ
Pure	BCP	0.718	0.034	−0.003	0.010	−0.021	−0.011	2.077	−0.323
	Al13-T	0.765	0.036	−0.030	0.011	−0.031	−0.019	2.659	−0.523
C	Al13-T	0.774	0.037	−0.043	0.012	−0.034	−0.023	2.907	−0.608
	Al14-T	0.735	-	-	-	-	-	-	-
N	Al13-T	0.781	0.037	−0.045	0.012	−0.035	−0.023	2.954	−0.624
	Al14-T	0.734	-	-	-	-	-	-	-
H	Al13-T	0.768	0.037	−0.042	0.012	−0.034	−0.022	2.903	−0.603
	Al14-T	0.769	0.037	−0.037	0.012	−0.033	−0.021	2.782	−0.572
O	Al13-T	0.787	0.038	−0.040	0.012	−0.034	−0.022	2.833	0.591
	Al14-T	0.770	0.039	−0.030	0.013	−0.034	−0.021	2.580	−0.523

Note: In the Al14-T region, after doping the I4 interstitial site with C or N, NNM disappears, but ELF_max_ still exists.

**Table 2 materials-13-02016-t002:** Octahedral interstitial atom formation energy at 0 shearing displacement.

Interstitial Site	Interstitial Formation Energy(eV/atom)			
	C	N	H	O
I1	−0.3887	−0.4190	−0.4016	−0.4660
I2	−0.3889	−0.4184	−0.4021	−0.4647
I3	−0.3892	−0.4188	−0.4015	−0.4646
I4	−0.4213	−0.4543	−0.4117	−0.4952
I5	−0.4192	−0.4496	−0.4113	−0.4887
I6	−0.4197	−0.4506	−0.4113	−0.4891

Note: The first column numbers I1–I6 correspond to the octahedral interstitial numbers in the model in Figure 1.

**Table 3 materials-13-02016-t003:** Unstable stacking fault energy (USFE) and intrinsic stacking fault energy (ISFE) of twin model.

	TiAl	TiAlC	TiAlN	TiAlH	TiAlO
USFE(mJ/m^2^)	496.07	498.52	561.20	511.53	657.08
ISFE(mJ/m^2^)	10.57	105.95	185.46	28.02	247.79

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
