# Peer review of "Influences of Multicenter Bonding and Interstitial Elements on Twinned γ-TiAl Crystal"

_materials, 2020, doi:10.3390/ma13092016_

Round 1
Reviewer 1 Report
- The abstract does not present the aim of work clearly.
- The introduction is unnecessarily lengthy and can be summarised.
- Although the theoretical model and results are quite well-presented, English language has to be modified.
Author Response
We thank editors for their whole-heartedly editing and the referees for their illuminating comments, which help us to refine our manuscript. We would like to respond to the comments from the referee as follows.
Reviewer:
Comments and Suggestions:
1) The abstract does not present the aim of work clearly.
2) The introduction is unnecessarily lengthy and can be summarised.
3) Although the theoretical model and results are quite well-presented, English language has to be modified.
Response:
Dear reviewer:
I am very sorry for our manuscript. According with your advices, we amend the relevant part in manuscript as following:
1) We modify the abstract to better match the aim of the work.
2) We revise the introduction.
3)We modify the English syntax and grammar.
Thanks again for your precious comments!

Reviewer 2 Report
Taking into account all the several features, the accuracy, scientific quality, scientific content and the interpretation of the results are interesting.
- The approach is interesting and the topic is appropriate for the journal.
- The work has a clear structure and all the sections are well written in a way that is easy to read and understand. In addition, the structure of the paper is good.
- The paper is focused on the three different kinds of interface relationships among γ/γ interface in polysynthetic twinned TiAl crystal, the proportion of true twin boundaries is the highest because it has the lowest interfacial energy, whose reason is discussed by local energy and three-center bond through first-principles calculations.
- It seems that the paper does not contain repetitions.
- The title is adequate and appropriate for the content of the article.
- The abstract contains information of the article.
- Figures and captions are essential and clearly reported.
Author Response
We thank editors for their whole-heartedly editing and the referees for their illuminating comments, which help us to refine our manuscript. We would like to respond to the comments from the referee as follows.
Reviewer:
Comments and Suggestions:
1) Taking into account all the several features, the accuracy, scientific quality, scientific content and the interpretation of the results are interesting.
2) The approach is interesting and the topic is appropriate for the journal.
3) The work has a clear structure and all the sections are well written in a way that is easy to read and understand. In addition, the structure of the paper is good.
4) The paper is focused on the three different kinds of interface relationships among γ/γ interface in polysynthetic twinned TiAl crystal, the proportion of true twin boundaries is the highest because it has the lowest interfacial energy, whose reason is discussed by local energy and three-center bond through first-principles calculations.
5) It seems that the paper does not contain repetitions.
6) The title is adequate and appropriate for the content of the article.
7) The abstract contains information of the article.
8) Figures and captions are essential and clearly reported.
Response:
I am very grateful to your comments for the manuscript. Your opinions are very important to us.

Reviewer 3 Report
The manuscript considers important aspect of gamma TiAl properties - its ductility. The authors in relatively broad way, described the known strategies allowing to reduce britleness of the above materials including control of microstructure and alloying.
My main objection to this manuscript is related with its form - in many places chaotic and messy written, e.g.
- p.5 "The projector augmented wave (PAW) method is used with the Perdew-Burke-Ernzerhof (PBE) generalized gradient approximation (GGA) functional in order to obtain the energy and stress densities..." there are no such a think like stress density in the DFT method,
- p.6 "The electron self-consistent energy convergence standard is 10-4 eV, and the cutoff used in the calculation is determined as 500 eV by the convergence test..." this 500eV "cutoff" is the max kinetic energy of plane waves used in VASP and PAW approach.
- The model of twin boundary involves only 12 atomic planes, taking into account periodic boundary conditions, there are no one, but two twin boundaries leading to one boundary per 6 atomic planes. It is very questionable whether such a small number of atomic planes is enough to eliminate boundary-boundary interaction which can influence on other calculated properties. Authors also did not mention about this issue and it seems there was no tests to check it.
- Figure 5: "Stack fault energy curve of twin γ-TiAl" should be generalised stacking fault energy curves. The horizontal axis cannot be strain, in GSFE calculations, displacements along the defined Burgers vector are analysed. Values on this axis are also unknown - what they indicate? What is the unit???
- The authors considered also slip scenario along atomic planes with interstitial atoms. In such case, the octahedral site is destroyed during the slip which enforces the interstitial atoms to move - this transition i.e. slip + transition of interstitial atom has to be modeled with the nudged elastic band method (NEB) as it is not possible to achieve reliable results without additional constraints during atomic relaxation. For more details see: Solid solution strengthening of hexagonal titanium alloys: Restoring forces and stacking faults calculated from first principles, Acta Materialia 102 (2016) 304.
Based on the above points and the general, very non-reader-friendly text body, I do not recommend this manuscript for publication.
Author Response
We thank editors for their whole-heartedly editing and the referees for their illuminating comments, which help us to refine our manuscript. We would like to respond to the comments from the referee as follows.
Reviewer:
Comments and Suggestions:
The manuscript considers important aspect of gamma TiAl properties - its ductility. The authors in relatively broad way, described the known strategies allowing to reduce britleness of the above materials including control of microstructure and alloying.
My main objection to this manuscript is related with its form - in many places chaotic and messy written, e.g.
1) p.5 "The projector augmented wave (PAW) method is used with the Perdew-Burke-Ernzerhof (PBE) generalized gradient approximation (GGA) functional in order to obtain the energy and stress densities..." there are no such a think like stress density in the DFT method,
2) p.6 "The electron self-consistent energy convergence standard is 10-4 eV, and the cutoff used in the calculation is determined as 500 eV by the convergence test..." this 500eV "cutoff" is the max kinetic energy of plane waves used in VASP and PAW approach.
3) The model of twin boundary involves only 12 atomic planes, taking into account periodic boundary conditions, there are no one, but two twin boundaries leading to one boundary per 6 atomic planes. It is very questionable whether such a small number of atomic planes is enough to eliminate boundary-boundary interaction which can influence on other calculated properties. Authors also did not mention about this issue and it seems there was no tests to check it.
4) Figure 5: "Stack fault energy curve of twin γ-TiAl" should be generalised stacking fault energy curves. The horizontal axis cannot be strain, in GSFE calculations, displacements along the defined Burgers vector are analysed. Values on this axis are also unknown - what they indicate? What is the unit???
5) The authors considered also slip scenario along atomic planes with interstitial atoms. In such case, the octahedral site is destroyed during the slip which enforces the interstitial atoms to move - this transition i.e. slip + transition of interstitial atom has to be modeled with the nudged elastic band method (NEB) as it is not possible to achieve reliable results without additional constraints during atomic relaxation. For more details see: Solid solution strengthening of hexagonal titanium alloys: Restoring forces and stacking faults calculated from first principles, Acta Materialia 102 (2016) 304.
Response:
I am very grateful to your comments on the manuscript. According with your advices, we amended the relevant part in manuscript as following:
1) The description mentioned is indeed for VASP. Energy density and stress density are in QMAS. We inserted to the wrong place the introduction of energy density and stress density. We are sorry about this
2) As the reviews mentioned, the description is for VASP instead of QMAS. We inserted to the wrong place. We are sorry about this
3) We did test the number of atomic layers of the twin model, and finally chose the model of 12 atomic layers. And when the twin model has 12 atomic layers, the layer energy of Layer4 and Layer10 is very close to the bulk model. Relative decription is amended to the manuscript.
4) We have modified the name and horizontal axis of Figure 5. We changed ‘Strain’ to ‘Displacement along 1/2[11-2]’.
5) < Solid solution strengthening of hexagonal titanium alloys: Restoring forces and stacking faults calculated from first principles, Acta Materialia 102 (2016) 304> is an excellent article, we will quote it in our article. Nudged elastic band method (NEB) is a very good method for studying the movement of atoms. We will consider this method in the follow-up work. But in this article, our focus is on the interaction between interstitial atoms and the interface, not the movement of interstitial atoms. And some other articles use the similar method to ours, such as: Effect of nitrogen on generalized stacking fault energy and stacking fault widths in high nitrogen steels, Acta Materialia 54 (2006) 2991.
Thanks again for your precious comments!

Reviewer 4 Report
1) First of all, the text in this paper needs to be revise by a native, English-speaking person. For example, the author expressed this paper in the first person.
2) The capital letters and the lower case letters are mixed up.
3) Also, the format of the manuscript is not organized, since the each section is not clear to understand.
4) Need the information for the sample and equipment. (model number, company name and country,etc.)
5) Please indicate clearly the final purpose of this study in the text.
Author Response
We thank editors for their whole-heartedly editing and the referees for their illuminating comments, which help us to refine our manuscript. We would like to respond to the comments from the referee as follows.
Reviewer:
Comments and Suggestions:
1) First of all, the text in this paper needs to be revise by a native, English-speaking person. For example, the author expressed this paper in the first person.
2) The capital letters and the lower case letters are mixed up.
3) Also, the format of the manuscript is not organized, since the each section is not clear to understand.
4) Need the information for the sample and equipment. (model number, company name and country,)
5) Please indicate clearly the final purpose of this study in the text.
Response:
I am very grateful to your comments for the manuscript. According with your advices, we amended the relevant part in manuscript as following.
1) We modify the English languages. Please check the change list
2) We check the spelling of the article. Please check the change list
3) We adjust the format of the article to make the each section easier to read.
4) We use Vienna Ab initio Simulation Package (VASP) code, CRITIC2 and QMAS in our calculations, and we explained in the article.
5) We explain the purpose of the article more clearly at the end of introduction.
Thanks again for your precious comments!

Reviewer 5 Report
The paper brings new data to explain the structure-property relationship in TiAl phase. This phase is of a great interest, because of its use in aerospace industry. I have no chance to check the calculations, but the results seem to be sound and correlate well with the up-to-date knowledge and experimental results.
The paper is not formatted using the standard journal's template, so it has to be re-formatted. The paper needs just minor language polishing.
Author Response
We thank editors for their whole-heartedly editing and the referees for their illuminating comments, which help us to refine our manuscript. We would like to respond to the comments from the referee as follows.
Reviewer:
Comments and Suggestions:
The paper brings new data to explain the structure-property relationship in TiAl phase. This phase is of a great interest, because of its use in aerospace industry. I have no chance to check the calculations, but the results seem to be sound and correlate well with the up-to-date knowledge and experimental results.
The paper is not formatted using the standard journal's template, so it has to be re-formatted. The paper needs just minor language polishing.
Response:
I am very grateful to your comments for the manuscript. According with your advices, we amended the relevant part in manuscript as following:
1) We modify the format of the article to match the journal
2) We modify the English languages of the article. Please check the change list
Thanks again for your precious comments!

Round 2
Reviewer 3 Report
The manuscript can be published in present form.
Author Response
I am very grateful to your positive comments for the manuscript. Your opinions are very important to us.
Reviewer 4 Report
Unfortunately, this manuscript should be revise due to following reasons.
1) The text body in the theoretical method and model should be divided by heading same as the result and discussion section, so that the contents will be easy to understand for the readers.
2) Line 82 in page 2 and lines 125 and 133 in page 3 should be fixed the form of the equations.
3) Line 160 in page 4, line 170, 173, 180 and 181, 185, 189, 190 and 192 in page 5 etc. should be changed.
4) I mentioned earlier, the text in this paper needs to be revise by a native, English-speaking person. For example, the author expressed this paper in the first person. However, they did not changed. (last part in the Introduction section) Also, please indicate clearly the final purpose of this study in the text.
Author Response
Reviewer:
Unfortunately, this manuscript should be revise due to following reasons.
1) The text body in the theoretical method and model should be divided by heading same as the result and discussion section, so that the contents will be easy to understand for the readers.
2) Line 82 in page 2 and lines 125 and 133 in page 3 should be fixed the form of the equations.
3) Line 160 in page 4, line 170, 173, 180 and 181, 185, 189, 190 and 192 in page 5 etc. should be changed.
4) I mentioned earlier, the text in this paper needs to be revise by a native, English-speaking person. For example, the author expressed this paper in the first person. However, they did not changed. (last part in the Introduction section) Also, please indicate clearly the final purpose of this study in the text.
Response:
Dear reviewer:
I am very sorry for our manuscript. According with your advices, we amend the relevant part in manuscript as following:
1) We divided ‘Theoretical method and model’ into ‘Model and calculation parameters’ and ‘Theoretical methods’
2) We modify the mentioned parts if our understanding on the comments is right.
3) We modify the unreasonable expression at lines you suggested.
4) About the language, we have asked the editor of the Press whose native language is English to make changes. And we also do our best to revise the English expression again since the given two days is not enough to hire a translator. We modify the narrative mode of the first person in the article if our understanding on the comments is right. We explain the purpose further in the last paragraph of ‘Introduction’.
Thank you very much again for your precious comments!
